# Drug Response Diversity: A Hidden Bacterium?

**DOI:** 10.3390/jpm11050345

**Published:** 2021-04-25

**Authors:** Nadji Hannachi, Laurence Camoin-Jau

**Affiliations:** 1Aix Marseille Univ, IRD, APHM, MEPHI, IHU Méditerranée Infection, 19-21 Boulevard Jean Moulin, 13005 Marseille, France; n_adji07@live.fr; 2Département de Pharmacie, Faculté de Médecine, Université Ferhat Abbas Sétif I, Sétif 19000, Algeria; 3Laboratoire d’Hématologie, Hôpital de la Timone, APHM, Boulevard Jean-Moulin, 13005 Marseille, France

**Keywords:** pharmacomicrobiomics, drug response, bacteria, intestinal microbiota

## Abstract

Interindividual heterogeneity in response to treatment is a real public health problem. It is a factor that can be responsible not only for ineffectiveness or fatal toxicity but also for hospitalization due to iatrogenic effects, thus increasing the cost of patient care. Several research teams have been interested in what may be at the origin of these phenomena, particularly at the genetic level and the basal activity of organs dedicated to the inactivation and elimination of drug molecules. Today, a new branch is being set up, explaining the enigmatic part that could not be explained before. Pharmacomicrobiomics attempts to investigate the interactions between bacteria, especially those in the gut, and drug response. In this review, we provide a state of the art on what this field has brought as new information and discuss the challenges that lie ahead to see the real application in clinical practice.

## 1. Introduction

The term pharmacomicrobiomics was first used in 2010 when a new branch was defined to understand differential responses between humans to several drugs based on microbiota [1]. Starting from the beginning, the approach of individualization of treatment is not recent. Indeed, this notion has been initiated first by considering the underlying dysfunctions, mainly the dosage adjustment or even the change of the molecule carried out during renal or hepatic failure. Subsequently, clinicians have adopted the “therapeutic drug monitoring” for certain drug molecules, for which a blood concentration–therapeutic efficacy correlation is well established, associated with significantly different concentrations between patients taking the same dose of the same treatment, as is the case with ciclosporin or tacrolimus, known to have a narrow therapeutic window [2,3].

The advent of genetic techniques and their progressive availability has made it possible to explain the number of differential responses to treatments, ranging from ineffectiveness to fatal toxicities. Further, tools have been developed for use in clinical practice to define the genetic status of patients by predicting their drug response [4,5]. This was of great interest and marked a major step forward in the approach to therapeutic individualization.

However, studies of genetic diversity could not explain all the interindividual treatment responses. It is currently estimated that pharmacogenomics could only explain 20 to 95% of these events [6]. This has prompted researchers to investigate what may be the cause of a distinct response to treatment, outside genetics and known underlying dysfunctions.

Today, the intestinal microbiota attracts much interest in an attempt to explain, in addition to other previously well-known aspects, the differences in responses to treatment. Indeed, during the absorption of a given drug, the latter is confronted with a widely diversified ecosystem of bacterial species, qualified by several authors as a metabolic organ [7,8,9]. The composition of the intestinal microbiota is subject to great differences between individuals, but also within the same organism over time, as it is constantly influenced by host-related factors, such as the immune system, as well as external factors, such as diet [10].

In this review, we present an overview of the available literature related to the issue of microbiota–drug interactions, from inactivation, activation or toxicity induced by the intestinal microbiota to its modulation by drugs. We also discuss the challenges that await this new research approach for better use in clinical practice.

## 2. Impact of Gut Microbiota on Drug Effect

The human body contains several microorganisms whose number is close to that of its whole cells, which is several trillion. The involvement of these microorganisms has been well established in the regulation of the immune system as well as the metabolism of polysaccharides and vitamins [11]. Moreover, certain species forming part of the intestinal microbiota have the enzymatic machinery necessary for the biotransformation of molecules found in the intestinal lumen, including drugs (Table 1). This biotransformation concerns, on one hand, the drug molecules ingested, before their absorption to gain the blood circulation, but also drugs, or their metabolites, which have followed a hepatic elimination via the bile. The result of this bacterial transformation of xenobiotics, which mainly consists of hydrolysis and reduction, unlike hepatic metabolism consisting of oxidation and conjugation reactions, gives metabolites that can be inactive, active or even toxic [8]. In addition, in addition to hydrolysis reactions, the intestinal microbiota is endowed with N-oxide cleavage, proteolysis, deconjugation, and others [12].

### 2.1. The Gut Microbiota Responsible for Drug Response

#### 2.1.1. Sulfasalazine

Sulfasalazine is a good, if not the first, example of considering the intestinal microbiota as a major step in the metabolism of a drug. In fact, bacterial azo-reductases make it possible to cleave sulfasalazine into its two active metabolites, sulfapyridine and 5-amine salicylic acid, with anti-inflammatory effects, effective in the treatment of ulcerative colitis, Crohn’s disease and rheumatoid arthritis (Figure 1A) [13]. The demonstration of these bacterial azo-reductases dates back to 1930 when researchers described their involvement in activating an antibacterial drug, Prontosil, which showed no effect in vitro [14].

#### 2.1.2. Warfarin

Since 1954, warfarin has been the first oral anticoagulant to prevent and treat thromboembolic complications in humans [15]. Warfarin is characterized by a marked interindividual response variability. Several studies have examined the genetic status of patients receiving a vitamin K inhibitor and have succeeded in identifying mutations predictive of treatment response, notably the VKORC1 and CYP2C9 mutations, genes encoding enzymes involved in the metabolism of vitamin K and coumarin vitamin K inhibitors, respectively [16]. On the other hand, 35% of patients with delayed responses to warfarin remain unexplained [17]. It is only recently that researchers have begun to explore the possible influence of the intestinal microbiota in response to warfarin. This idea arose because of the known link between the gut microbiota and vitamin K metabolism. Recently, Wang et al. explored gut microbial diversity in 200 patients with a high, normal, or low response to warfarin. A significant presence of the genera *Bacteroides*, *Escherichia–Shigella* and *Klebsiella*, were reported in patients with a weak response, as the *Escherichia–Shigella* genera has the enzymatic machinery essential for synthesizing vitamin K, while that the genus *Enterococcus* was associated with an elevated response to warfarin (Figure 1B) [18]. This study is the first to establish a link between gut microbiota and response to warfarin, a narrow-spectrum therapeutic molecule, for which an unsuitable dosage can be associated with hemorrhage or, rather, in inefficiency. Such promising results need to be confirmed and refined by further research in the future.

#### 2.1.3. Digoxin

Digoxin is a natural cardiotonic glycoside that has been used for more than two centuries in the treatment of heart failure and certain types of arrhythmias. Digoxin is ineffective in 10 to 15% of patients treated with conventional doses [19]. The particularity of digoxin is that it is associated with a particular intestinal bacterium, *Eggerthella lenta*, originally classified as *Eubacterium lentum* (Figure 1C) [20]. It is noteworthy that this species shows increased growth in diabetic patients [21,22]. Although the mechanism of interaction is not clearly understood, strains belonging to this bacterial species are thought to possess a two-gene cytochrome encoding operon system, designed as a two-gene cardiac glycoside reductase (*cgr*) operon, which is significantly upregulated in the presence of digoxin, which in turn results in the reduction of digoxin to its inactive metabolite, dihydro-digoxin, in which the lactone ring is reduced [23,24]. It has been reported that the presence of *Eggerthella lenta* strains was associated with decreased efficacy of digoxin [24,25]. Importantly, arginine, a semi-essential amino acid for humans, shows inhibition of cgr, which results in the prevention of digoxin inactivation by *E. lenta*. Therefore, it is now assumed that a diet rich in amino acids, particularly arginine, may be conducive to an adequate response to digoxin treatment [23]. A peculiarity of the cgr system is that it has pockets, which bind both digoxin and other compounds, referred to as digoxin-like, as in the case of fumarate, which has a higher affinity than digoxin [26]. This particularity suggests the future development of adjuvant drugs preventing the activation of digoxin by this mechanism.

#### 2.1.4. Levodopa

Species of the gut microbiota, which are commensal and normally free from any adverse effects, are not the only ones to have a well-established effect on optimal drug activity. Indeed, it has been shown that the presence of *Helicobacter pylori*, the bacterium responsible for gastric ulcers, was significantly associated with a decrease in levodopa absorption, the benchmark drug for Parkinson’s disease (Figure 1D) [27]. Meanwhile, an increase in levodopa absorption of about 21 to 54% was obtained following the eradication of this bacterium [28]. The supposed mechanism of interaction is linked to the modification of gastric pH caused by the bacteria. Another hypothesis suggests a physical attachment between the bacterium and the drug leading to a decrease in the bioavailability [28,29]. In addition to levodopa, helicobacter pylori also appear to decrease the absorption of thyroxine, delavirdine and iron supplements [30].

#### 2.1.5. Chemotherapy and Immunomodulator Drugs

It is not surprising that the intestinal microbiota can influence the response to immunomodulating and cytotoxic treatments, as long as it is directly related to the functioning of the immune system [31,32]. In addition, patients receiving immunomodulating and cytotoxic drugs, mainly the cancer population, already have multifactorial impairment of the intestinal microbiota by the host environment and diet, surgery, using adjuvant drugs, as well as by the effect of these drugs themselves.

The intestinal microbiota influences the action of these drug classes via xenometabolism and community structure alteration mechanisms and via immunomodulation mechanisms [33]. These interactions can occur either intraluminal or in the lymphoid organs following drug-induced bacterial translocation [34,35]. In this context, one study reported that cyclophosphamide (CTX), an alkylating drug whose function also depends on the stimulation of anticancer immunity, induced transmucosal translocation of specific bacteria, such as *Enterococcus hirae* and species belonging to the *Lactobacillus* genus (*Lactobacillus johnsonii*, *Lactobacillus murinus*), in the mesenteric lymph nodes and the spleen, which lead to T-helper 17 (Th17) cell differentiation, thereby enhancing the adaptive antitumor immune response to CTX [36]. In another study, Daillère et al. have demonstrated that oral gavage of *E. hirae* in antibiotic pretreated mice could restore response to CTX (Figure 2A) [37]. In addition, it has been reported that a memory Th1 immune response towards *E. hirae* may predict progression-free survival in patients with end-stage lung and ovarian cancer [33,38].

Another association exists, this time with immune checkpoint inhibitors and *Bacteroides* species. Indeed, T-cell reactions to *Bacteroides fragilis* have been found in patients with melanomas responding to treatment with CTLA-4 checkpoint inhibitors. In vivo investigations were able to restore the response to ipilimumab in germ-free mice by administering *B. fragilis* or adoptive transfer *B. fragilis*-specific T cells [39].

Several studies have also associated the abundance of *Akkermentia muciniphila*, *Collinsella aerofaciens Enterococcus faecium*, Ruminococcaceae family and *Bifidobacterium* spp. with an adequate response to Anti PD-1 [40,41,42]. Moreover, fecal microbial transplantation of human responders into germ-free mice restored the antitumor effect of PD-1 blockade in the recipient mice. Another study reported an interaction between *Bifidobacterium* and dendritic cells, resulting in T cells activation and enhancement of the protective anticancer response of anti-PD-L1 [43].

In addition to the intestinal microbiota, bacteria can also modulate the effect of a drug while localizing in the tumor tissue. This is the case of *Mycoplasma hyorhinis*, which has enzymes catabolizing nucleoside analogs, making it responsible for gemcitabine resistance. Other bacteria, especially those belonging to the Gammaproteobacteria, are also responsible for gemcitabine resistance (Figure 2B). Interestingly, a significant percentage of ductal adenocarcinomas of the human pancreas, a type of tumor commonly treated with gemcitabine, contain the culprit bacteria [44]. Interestingly, the administration of ciprofloxacin to rodents could reverse the gemcitabine resistance induced by intratumoral Gammaproteobacteria. In another study, Lehouritis et al. reported that gemcitabine efficacy might also be impaired by *E. coli* [45].

Other cytotoxic molecules subject to the influence of the intestinal microbiota, but in a different context, will be discussed below.

### 2.2. The Gut Microbiota Causing Drug Toxicity

#### 2.2.1. Irinotecan

The gut microbiota can be the cause of drug toxicity. The well-established link between the gut microbiota and drug-related toxicity, to date, is bacterial beta-glucosidase. These β-glucosidases allow the hydrolysis of the hepatic glycoside metabolites, called aglycones. Indeed, it has been reported by various studies that several drugs, or classes of drugs, could be the substrate of these enzymes giving rise to the formation of toxic metabolites. The first example is camptothecin-11 (CPT-11), also known as irinotecan, a drug used to treat colon cancer. Indeed, CPT-11 is a prodrug activated initially by hepatic carboxylesterases, giving rise to SN-38, responsible for the cytotoxic effect. In the second step, the SN-38 will be glucuronidased also at the hepatic level to obtain the SN-38G so that it is excreted via the bile to reach the intestine. At this level, bacterial β-glucuronidases will make the opposite reaction, reconverting SN-38G into SN-38, toxic for the intestinal epithelial cells, causing intense diarrhea in up to 80% of treated patients (Figure 3A) [46]. It should be noted that ciprofloxacin and low doses of amoxapine are effective in suppressing the bacterial activity of β-glucuronidase and mucositis due to the absorption of irinotecan [47,48]. Moreover, the search for specific inhibitors of these β-glucosidases seems to be an intriguing approach. Indeed, Cheng et al. reported that compound TCH-3562 showed specific inhibition of *E. coli* beta-glucosidases without affecting human β-glucosidases [49].

Toxic effects of chemotherapy drugs can also occur because of the bacterial metabolism of another drug taken concomitantly. This is reminiscent of the story of the 16 Japanese cancer patients treated simultaneously with 5 fluorouracil and sorivudine, a potent antiviral drug, where the intestinal microbiota was the first culprit [50]. Indeed, enzymes from the *Bacteroides* genus, such as *Bacteroides vulgatus*, *B. thetaiotaomicron* and *Bacteroides eggerthii*, metabolized sorivudine into bromovinyluracil, a metabolite that inactivates hepatic dihydropyrimidine dehydrogenase, an enzyme involved in the inactivation of 5 fluorouracil [51,52]. This resulted in extremely high concentrations of 5 fluorouracil, inducing death (Figure 3B) [53].

#### 2.2.2. Nonsteroidal Anti-Inflammatory Drugs

As with Irinotecan and through the same bacterial glucuronidases, nonsteroidal anti-inflammatory drugs (NSAIDs) are metabolized in the gut to give toxic metabolites to the intestinal mucosa. In fact, in addition to the gastric ulcers induced by this drug class via the inhibition of the synthesis of protective prostaglandins of the gastric wall, it has been described, with the advent of new gastroenterological exploration techniques, that NSAIDs caused mucosal damage in the small intestine [54]. Indeed, the NSAIDs glucuronides by the liver reach the intestine via the bile. At this level, the bacterial β-glucuronidase hydrolyze them into aglycones, which are again reabsorbed and taken in charge by the cytochrome P450 to give potentially cytotoxic intermediates, responsible for this intestinal toxicity (Figure 3C) [55]. Like TCH-3562 for irinotecan, a recent study reported that Inh1 showed a reduction in the intestinal side effects of diclofenac via specific inhibition of β-glucosidases in an animal model [56].

Although several commensal bacteria of the intestine produce β-glucuronidases of different sequences and structures, providing beneficial functions to the organism, it has been reported that de-glucuronidation of drugs, leading to toxic metabolites, is carried out mainly by opportunistic or enteropathogenic bacteria, in particular, *Clostridium perfrengens* and *Escherichia coli*. This particularity is explained by internal differences between the different types of this enzyme, differences concerning conformations, hydrophobicity and flexibility [57]. The potentiating effect of bacterial β-glucosidase-induced drug toxicity, particularly intestinal toxicity, is not limited to irinotecan and NSAIDs. Other molecules have also been shown to be substrates for these enzymes, such as Regorafenib, a tyrosine kinase inhibitor with an antitumor effect, as well as venotonic flavonoids [58,59,60].

#### 2.2.3. Impact of Non-Antibiotic Drugs on the Gut Microbiota

In the other direction, the gut microbiota is not spared from disturbance or influence by drugs. It is assumed that 10% of the interindividual variations of the gut flora composition can be explained by drug use [7,61]. Indeed, several drugs are known to induce dysbiosis. Others push the growth of particular bacterial species resulting in a beneficial effect in humans.

#### 2.2.4. Proton Pomp Inhibitors

Proton pump inhibitors (PPI) decrease gastric acidity and are used primarily to treat ulcers and gastritis. Through the increase in gastric pH, the bacteria present in the oral cavity find the capacity to release and maintain themselves in the stomach and gut [62,63]. In addition, by this decrease in acidity, pathogenic bacteria using the oral route also find this barrier weakened, as are the cases of *Clostridium difficile* responsible for pseudomembranous diarrhea as well as *Salmonella* and diarrheagenic *Escherichia coli* (Figure 4A) [64,65]. In addition, other studies have associated using PPIs with a decrease in the abundance of certain commensal bacteria in the gut, such as *Bifidobacterium* spp. and *A. muciniphila*, versus an increase in β-glucuronidase-producing bacteria [66,67,68].

#### 2.2.5. Metformin

Metformin is an antihyperglycemic agent used as a first-line treatment in type 2 diabetics. It is also used in obese people to reduce fat mass. Intravenous metformin administration has been shown to be associated with reduced blood-glucose-lowering relative to oral metformin [59]. Indeed, the mechanism of action of metformin is based on the decrease in hepatic synthesis and intestinal absorption of glucose and an increase in the sensitivity of muscle cells to insulin [69]. Metformin has been reported to induce changes in the composition of the intestinal flora, making it rich in bacteria like *Roseburia* and *butyrivibrio* genera producing short-chain fatty acids (SCFAs), such as butyrates. These SCFAs boost glycolysis, enhance epithelial barrier function by promoting epithelial growth and immune responses to damage [70]. Metformin also induces bacteria degrading mucin-like *A. muciniphila*, which could mediate the therapeutic effect of metformin by promoting intestinal stem cells-mediated epithelial development contributing thus to maintain intestinal homeostasis (Figure 4B) [71,72,73,74]. It is important to note that these species are found below normal in diabetic patients [75,76]. The use of metformin thus restores better epithelial permeability and improves glucose and lipid metabolism.

## 3. Gut Microbiota—Drug Bidirectional Interaction

### Methotrexate

Another example of the modulation of the composition of gut microbiota promoting therapeutic drug effect is that of methotrexate (MTX). MTX is a cytotoxic folate analog. It inhibits dihydrofolate reductase and thymidylate synthase, preventing *de novo* pyrimidine and purine synthesis. It is mainly used in cancer and the treatment of autoimmune diseases, such as rheumatoid arthritis and psoriasis. MTX decreased the intestinal abundance of the enterobacterial group, especially species most closely related to *Enterococcus faecium* [77]. This could be beneficial for patients receiving MTX, as these species are disease-associated bacteria associated with increased intestinal permeability and induced inflammatory process. Other bacteria are also influenced by this drug, such as an increase in the *Lachnospiraceae* family and a decrease in Ruminococcaceae, Bacteroidetes phyla and *Bacteroides fragilis*, which could, this time, accentuate the gastrointestinal side effects of the drug, particularly intestinal mucositis [78,79]. It is worth noting that MTX, taken orally in humans, modifies the composition of the dental and salivary microbiota more than that of the intestinal microbiota [80,81]. The MTX-gut microbiota interaction is not unidirectional. Indeed, MTX appears to be more influenced by the intestinal microbiota than the opposite. In this sense, it has been shown that the diversity of the gut microbiota can determine the response to the treatment. Zhang et al. reported a correlation between the diversity of the digestive flora with the response to MTX after 3 months of treatment, with a statistically superior abundance of *Prevotella maculosa* in responders, compared to nonresponder patients [81].

Globally, MTX has highly variable interindividual bioavailability, ranging from 10% to 80%, with only 40% of patients achieving therapeutic blood concentration [82,83]. In addition, there is a narrow therapeutic range and numerous side effects, including nephrotoxicity, hepatotoxicity, gastrointestinal and hematological toxicities. Several teams have focused on the search for genetic factors predicting the response to MTX [84], others have proposed dosing MTX polyglutamate in red blood cells, but all of these approaches have not achieved clinical relevance [85,86]. This resulted in particular interest in the search for signatures of the intestinal microbiota to predict the response to this drug. The results of studies combining methotrexate and digestive flora discussed above open up promising prospects for achieving this goal. Further studies are required to assess the feasibility and allow its introduction into clinical practice.

## 4. Treatment Failure with TNF Alpha Inhibitors Due to Bacteria

Antitumor necrosis factor-alpha (anti TNF α) is used to treat autoimmune diseases, such as rheumatoid arthritis and Crohn’s disease. Specifically, in some patients diagnosed with rheumatoid arthritis, anti-TNF alpha antibodies show no improvement but instead exacerbate the disease. Several case reports have associated this therapeutic failure with the presence of a particular intracellular bacterium called *Tropherima whipplei* [87,88,89]. In fact, this bacterium can be carried by healthy subjects and is mainly found in stool and saliva, in which it does not induce any clinical expression. However, in some patients, it is responsible for a disease called Whipple’s disease, which is a group of gastrointestinal disorders with joint pain [90,91]. It is precisely because the onset of this disease is triggered by joint pain that it can be confused with rheumatoid arthritis and explains using anti-TNF α drug, which accentuates the infection. It should be noted that this bacterium can also cause infectious endocarditis [92]. Today several researchers are suggesting the prior search for *T. whipplei* in patients diagnosed with rheumatoid arthritis and programmed to receive anti-TNF α [93,94].

## 5. Implementation of Pharmacomicrobiomics in Clinical Practice

Although there is growing evidence of the involvement of the intestinal microbiota in drug response, consideration of this question is not part of routine clinical practice. Indeed, its use could be of great interest, involving microbiologists, pharmacologists and especially clinical pharmacists in the choice and optimization of treatments. First, studies are needed to determine the best probiotics and prebiotics, which could promote the management of a certain class of drugs, such as methotrexate, by restoring the intestinal flora, which is essential to the efficacy of the drug. Second, the benefit of antibiotic use with certain chemotherapy molecules needs to be better determined, such as the use or not of ciprofloxacin in patients treated with gemcitabine or irinotecan, or even by using specific inhibitors of bacterial enzymes, such as beta-glucosidases. In addition, understanding the interactions between the intestinal microbiota and drugs could make it possible to avoid the association of certain drugs, the first one increasing or decreasing the abundance of one bacterium involved in the effectiveness of the other. Fecal microbiota transplantation (FMT), initially used in the non-drug treatment of *C. diffcile* pseudomembranous colitis and ulcerative colitis [95,96], may also show a benefit in the management of drug efficacy. Indeed, FMT is beginning to show promising results in the management of immune checkpoint inhibitors as well as in the therapeutic regimen with FOLFOX, a 5-Fluorouracil, leucovorin and oxaliplatin-based cocktails used in the treatment of colorectal cancer [97,98]. However, it should be noted that this technique currently requires to be studied further to become more standardized [99,100]. Finally, the prior search for particular bacterial species or specific strains whose link with the response to a given treatment needs to be better established, as in the case of *E. lenta* with digoxin, *H. pylori* with levodopa, or the search for *T. whipplei* in patients programmed to receive anti-TNF α, which would allow to choose or avoid a drug molecule from the outset without falling into ineffectiveness or rather toxicity.

However, this introduction could not be tangible without additional effort by the researchers. To date, the techniques used to study the intestinal microbiota in relation to responses to treatment or for other purposes are mainly based on metagenomics, metabolomics, culturomics and bioinformatics. These techniques can provide considerable assistance in developing sensitive, specific and relatively rapid tests to detect particular species, unlike mass analysis. This could be translated by developing selective culture media, Q-PCR kits or even searching for a special metabolite that indirectly reveals the presence of the bacteria being tested for. Immunological techniques may also be interesting in the search for specific antigens, antibodies or even in searching for specific T-lymphocytes of the bacteria searched as it was initiated for *B. fragilis* in the treatment with ipilimumab. Further investigations are needed to characterize the specific genes involved in such mechanisms, which could ultimately assist to rise the quest for bacterial-induced drug modulation.

## Figures and Tables

**Figure 1 jpm-11-00345-f001:**
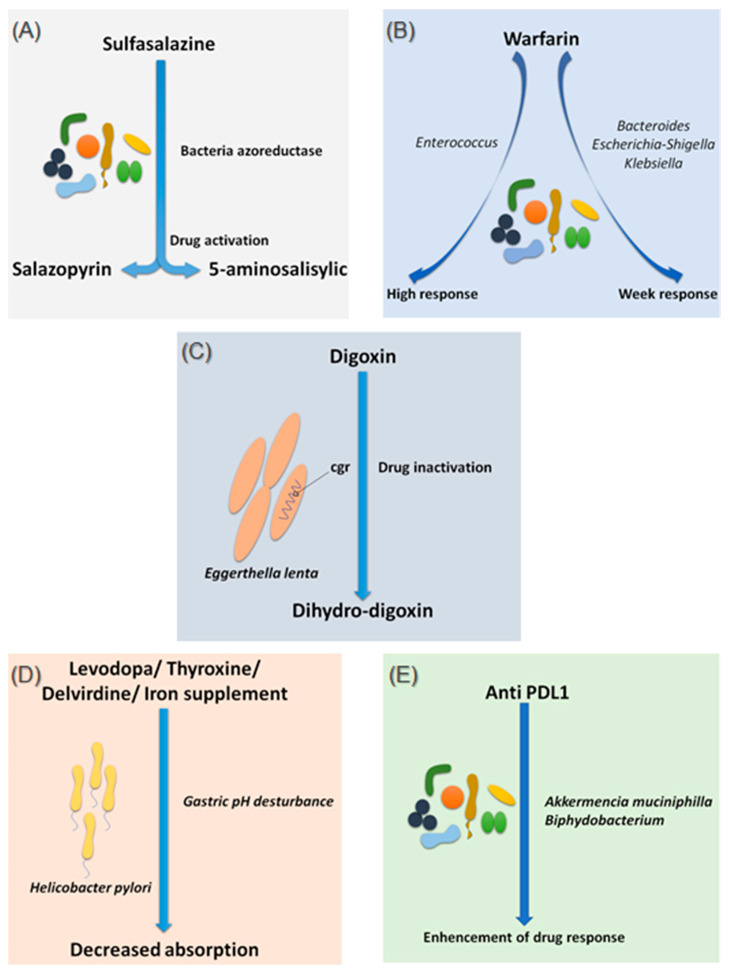
Effects of bacteria on drug absorption and response to treatment. (**A**) Effect of bacterial azo reductases on the bioactivation of sulfasalazine. (**B**) Effect of the different compositions of the intestinal microbiota on the response to warfarin. (**C**) Effect of *Eggerthella lenta* bacteria on the bio-inactivation of digoxin. (**D**) Involvement of *Helicobacter pylori* in gastric pH modification leading to decreased drug absorption. (**E**) Involvement of *Akkermansia muciniphila* and *Bifidobacterium* in improving the response to anti-PDL1 treatments. Illustrations were created partially with biorender.com.

**Figure 2 jpm-11-00345-f002:**
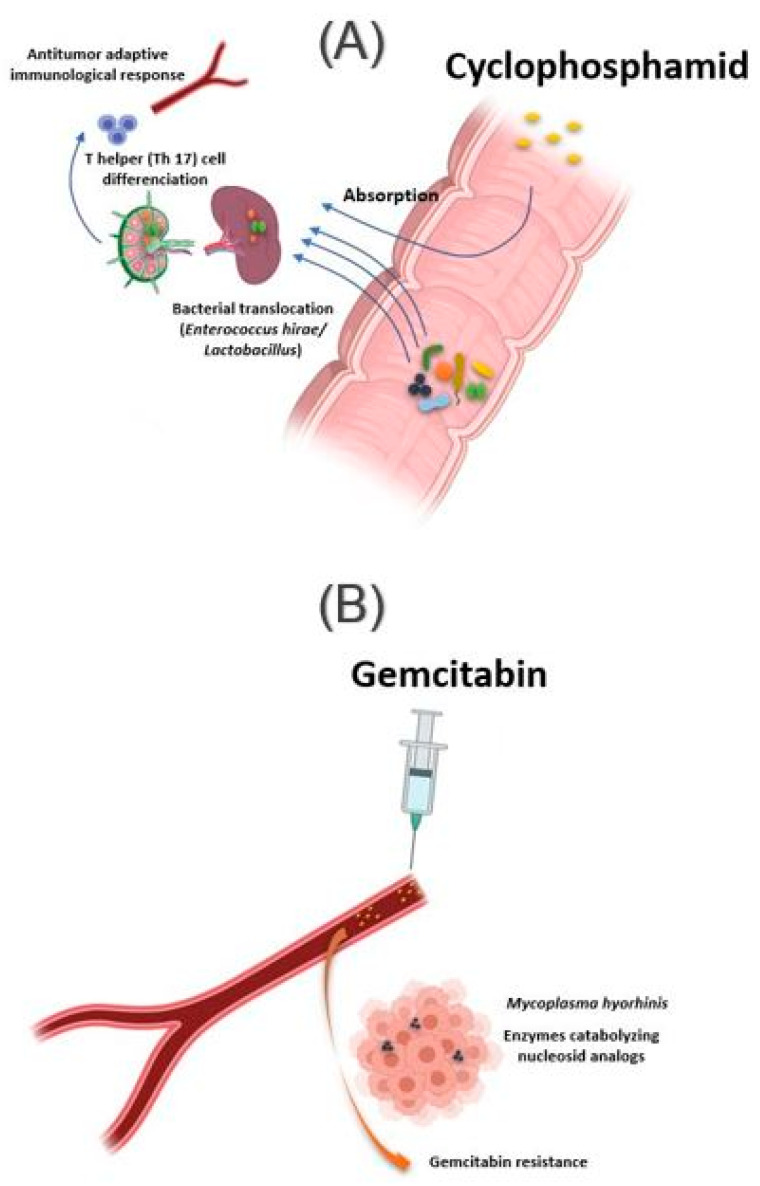
Influence of bacteria found in tissue on response to cancer chemotherapy treatment. (**A**) Oral absorption of cyclophosphamide induces transmucosal translocation of specific bacteria, such as *Enterococcus hirae* and species belonging to the *Lactobacillus* genus (*Lactobacillus johnsonii*, *Lactobacillus murinus*), in the mesenteric lymph nodes and the spleen. This leads to T-helper 17 (Th17) cell differentiation, enhancing the adaptive antitumor immune response to CTX. (**B**) The presence of Mycoplasma hominis in the tumor leads to resistance to gemcitabine treatment via their enzymes catabolizing nucleoside analogous. Illustrations were created partially with biorender.com.

**Figure 3 jpm-11-00345-f003:**
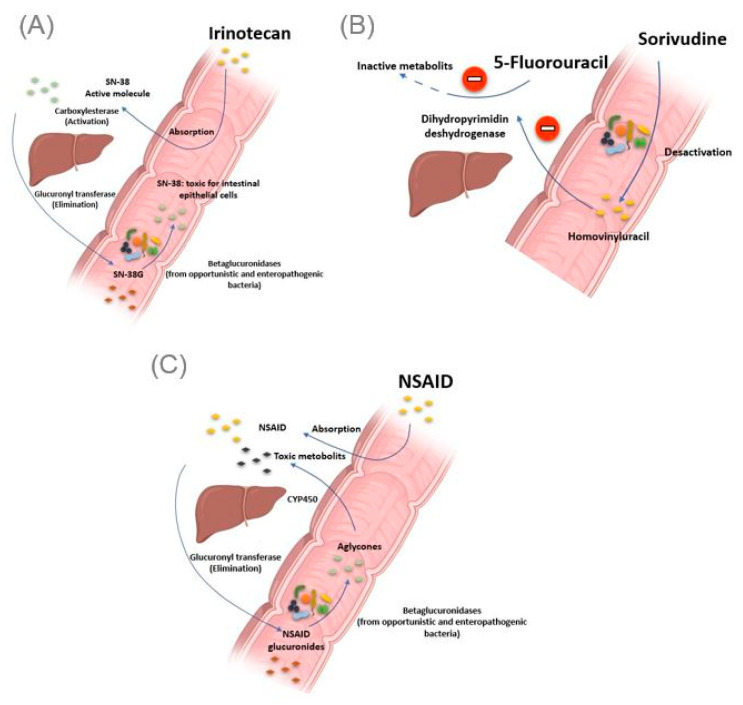
Intestinal bacteria involved in drug toxicity. (**A**) The prodrug irinotecan is activated initially by hepatic carboxylesterases, giving rise to SN-38, responsible for the cytotoxic effect. Second, the SN-38 is glucuronidased in the liver to obtain the SN-38G, which is excreted via the bile to reach the intestine. At this level, bacterial β-glucuronidases make the opposite reaction, reconverting SN-38G into SN-38, toxic for the intestinal epithelial cells, causing intense diarrhea. (**B**) Bacterial enzymes metabolize sorivudine into bromovinyluracil. The latter is absorbed and inactivates hepatic dihydropyrimidine dehydrogenase, an enzyme involved in the inactivation of 5 fluorouracil. This resulted in extremely high concentrations of 5 fluorouracil, inducing death. (**C**). NSAIDs are normally glucuronidated in the liver. NSAIDs glucuronides reach the intestine via the bile. At this level, the bacterial beta-glucuronidase hydrolyzes them into aglycones, which are again reabsorbed and taken in charge by the cytochrome P450 to give potentially cytotoxic intermediates responsible for intestinal toxicity. Illustrations were created partially with biorender.com.

**Figure 4 jpm-11-00345-f004:**
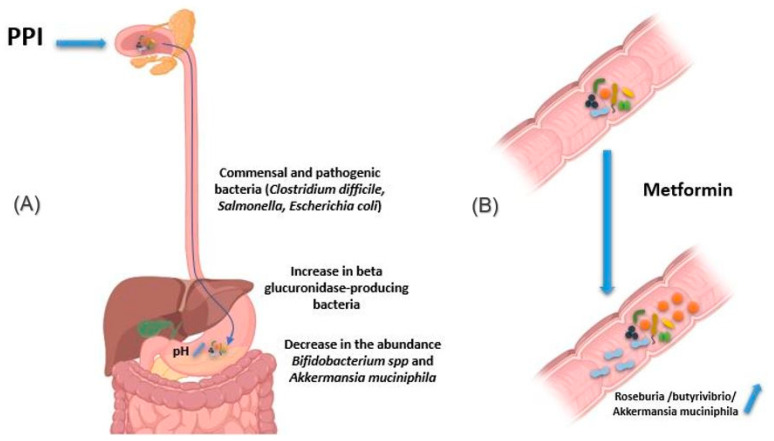
Effect of drugs in modifying the composition of the intestinal microbiota. (**A**) PPIs induce an increase in gastric pH. Thus, the bacteria present in the oral cavity find the capacity to release and maintain themselves in the stomach and gut. In addition, pathogenic bacteria using the oral route also find this barrier weakened, as are the cases of *Clostridium difficile*, *Salmonella* and diarrheagenic *Escherichia.* In addition, using PPIs is associated with a decrease in the abundance of certain commensal bacteria in the gut, such as *Bifidobacterium* spp. and *A. muciniphila*, versus an increase in beta glucuronidase-producing bacteria. (**B**) Metformin induces changes in the composition of the intestinal flora, making it rich in bacteria producing short-chain fatty acids, such as butyrates like *Roseburia* and *butyrivibrio* genera, and bacteria degrading mucin-like *A. muciniphila*. The use of metformin thus restores better epithelial permeability and improves glucose and lipid metabolism. Illustrations were created partially with biorender.com.

**Table 1 jpm-11-00345-t001:** Summary table on the different bacteria–drug interactions and their consequences.

Drugs	Microbs	Body Site	Effects	References
**Drug Effect Influenced by Bacteria**				
Sulfasalazine	Bacteria possessing azoreductase enzymes	Gut	Cleavage into its two active metabolites, Salazopyrin and 5-amine salicylic acid	Peppercorn MA and Goldman P, 1972
Warfarin	*Bacteroides*, *Escherichia–Shigella* and *Klebsiella*	Gut	weak response to the drug	Wang L et al., 2020
	*Enterococcus*	Gut	High response to the drug	
Digoxin	*Eggerthella lenta*	Gut	reduction of digoxin to its inactive metabolite, dihydro-digoxin	Haiser HJ et al., 2014; Koppel N et al., 2018
Levodopa	*Helicobacter pylori*	Stomach	decreased drug absorption	Hashim H et al., 2014
Cyclophosphamide (CTX)	*Enterococcus hirae*, *Lactobacillus johnsonii*, *Lactobacillus murinus*	Mesenteric lymph nodes and the spleen	Enhancement of the antitumor adaptive immunological response to CTX	Viaud S et al., 2013; Daillère et al., 2016
CTLA-4 checkpoint inhibitors	*Bacteroides fragilis*	Gut	Restore the response to the treatment	Vétizou M et al., 2015
Anti PD-1	*Akkermentia muciniphila*, *Collinsella aerofaciens*, *Enterococcus faecium*, Ruminococcaceae family, *Bifidobacterium* spp.	Gut	Enhanced response to treatment	Gopalakrishnan V et al., 2018; Matson V et al., 2018; Routy B et al., 2017
Gemcitabine	*Mycoplasma hyorhinis*, bacteria belonging to the Gammaproteobacteria, *Escherichia coli*	Tumor tissue	Gemcitabine resistance	Galler et al., 2017; Lehouritis P et al., 2015
Irinotecan	Opportunistic or enterohepatic bacteria possessing β-glucuronidases enzymes	Gut	Production of toxic metabolites responsible for diarrhea	Stein A et al., 2010
NSAIDs		Gut	Production of toxic metabolites responsible for mucosal damage in the small intestine	Higuchi et al., 2009; Boelsterli UA et al., 2013
**Bacteria abundance influenced by drugs**				
Proton pump inhibitors	*Clostridium difficile*, *Salmonella*, diarrheagenic *Escherichia coli* and beta glucuronidase-producing bacteria	Gut	Increased bacteria	Dial et al., 2004; Bruno G et al., 2019; Blackler RW et al., 2015; Davis JA et al., 2020; Wallace JL et al., 2011
	*Bifidobacterium* spp. and *Akkermentia muciniphila*	Gut	Decreased bacteria	
Metformin	*Roseburia*, *butyrivibrio* genera and *Akkermentia muciniphila*		Increased bacteria, responsible for better epithelial permeability and improvement in glucose and lipid metabolism	Forslund K et al., 2015; Shin NR et al., 2014; Wu H et al., 2017
**Bidirectional effect**				
Methotrexate (MTX)	*Enterobacterial* group, Ruminococcaceae, Bacteroidetes phyla and *Bacteroides fragilis*	Gut	Decreased bacteria	Ramos-Romero S et al., 2018; Zhou B et al., 2018
	Lachnospiraceae family	Gut	Increased bacteria	
	*Prevotella maculosa*	Gut	Enhancement of the response to the treatment	Zhang et al., 2015

## Data Availability

Not applicable.

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
