# Peer review of "Drug Response Diversity: A Hidden Bacterium?"

_jpm, 2021, doi:10.3390/jpm11050345_

Round 1

Reviewer 1 Report

The subject of the publication is interesting, useful and properly developed.
The introduction is clearly written, the whole is supported by properly selected literature.
Please correct typos, eg "131 - Chemotherapy and immunomodulator dugs".
I consider the article important from the medical point of view and
worth reading and attention.

Author Response

The subject of the publication is interesting, useful and properly developed.
The introduction is clearly written, the whole is supported by properly selected literature.

Please correct typos, eg "131 - Chemotherapy and immunomodulator dugs".
I consider the article important from the medical point of view and worth reading and attention.

Response:

Dear reviewer, we thank you for your comments.

As requested, spelling mistakes have been corrected. Please check the new submitted version.

Reviewer 2 Report

Major comments

  1. Line 51: clarify meaning here of “latter” – do you mean the drug – but this does not form an organ together with the microbiome
  2. Line 70: bacteria perform many other metabolism reactions besides hydrolysis. The majority is thought to be hydrolysis and reduction, but they are also know to perform functional group removal, N-oxide cleavage, proteolysis, denitration, deconjugation, amine formation/amide hydrolysis, and others.
  3. Digoxin section: needs a sentence or two describing the clinical significance of the presence of the particular strain of E. lenta on digoxin bioavailability and therapeutic effect.
  4. Impact of drugs on the gut microbiota: would be worth mentioning the role of antibiotics as one of the most profound mediators of the microbiota. Or could change the subtitle to “Impact of non-antibiotic drugs on the gut microbiota”
  5. Metformin section: need to describe how metformin’s therapeutic effect is mediated by these bacteria. Figure 4B does not add to the description in the text.
  6. Line 265: MTX alters salivary microbiota more than intestinal – has this been found in humans?  Needs a reference.
  7. Implementation of Pharmacomicrobiomics in Clinical Practice: this paragraph needs some re-working as we are not yet in a position to begin recommending specific probiotics, prebiotics, postbiotics or organisms in order to manipulate the gut microbiome to improve drug efficacy/toxicity. The review is a good summary of data to date – showing that there is definitely an important role for the gut microbiome in the mechanisms for altering drug efficacy/toxicity, but the field is still in its infancy – and exact molecular mechanisms area still elusive.  We can hypothesize for a day when we may be able to manipulate the gut microbes to assist in improving bioavailability and/or decrease toxicity; however, it is not accurate to suggest that adding or subtracting the specific organisms that have been discussed here is ready for clinical prime time in humans.

Minor comments

  1. Line 32: reword sentence; break into 2 sentences
  2. Line 35: change to “therapeutic drug monitoring”
  3. Lines 52-55: need references for these statements
  4. Line 108: “species shows increased growth in diabetic  patients” needs reference
  5. Line 115: change “conductive” to “conducive”
  6. Line 139: change to “xenometabolism”
  7. Line 188: change to “intense diarrhea in up to 80%
  8. Line 213: add “in an animal model” after b-glucosidases
  9. Line 227: reference the statement: “10% of inter-individual variations of gut flora composition can be explained by drug use.”
  10. Line 257: change “cancerology” to “cancer”
  11. Line 273: Rephrase 1st sentence – incomplete sentence
  12. Line 279: change “approaches have not been pursued” to “approaches have nott achieved clinical relevance”
  13. Line 284: consider changing section title to “Treatment failure with TNF alpha inhibitors due to bacteria”
  14. Line 286: (anti-TNFa ) is used…

Author Response

  1. Line 51: clarify meaning here of “latter” – do you mean the drug – but this does not form an organ together with the microbiome

Response :

Dear reviewer, the term "the latter" refers to drug. The term « organ » refers to the microbiota. This qualification has already been used by several authors before. As requested, the sentence in the manuscript has been clarified and referenced in the new submitted version. Please check the new submitted version: line 52; page 3.

  1. Line 70: bacteria perform many other metabolism reactions besides hydrolysis. The majority is thought to be hydrolysis and reduction, but they are also know to perform functional group removal, N-oxide cleavage, proteolysis, denitration, deconjugation, amine formation/amide hydrolysis, and others.

Response :

Dear reviewer, we totally agree. As requested, this point has been added and referenced. Please check the new submitted version: lines 69-73; page 4.

  1. Digoxin section: needs a sentence or two describing the clinical significance of the presence of the particular strain of E. lenta on digoxin bioavailability and therapeutic effect.

Response :

As requested, A sentence has been added regarding efficacy of digoxin in presence of eggerthella lenta strain. Please check the new submitted version: lines 125 and 126; page 7.

  1. Impact of drugs on the gut microbiota: would be worth mentioning the role of antibiotics as one of the most profound mediators of the microbiota. Or could change the subtitle to “Impact of non-antibiotic drugs on the gut microbiota”

Response :

In this section, we are mainly interested in non-antibiotic drugs. Therefore we have changed the title as suggested. Please check the new submitted version: line 264; page 13.

  1. Metformin section: need to describe how metformin’s therapeutic effect is mediated by these bacteria. Figure 4B does not add to the description in the text.

Response :

As requested, the effect of metformin on the gut microbiota was better explained in the new submitted version.  Please check the new submitted version: lines 300-305; page 15. Figure 4 B presents only a simplified diagram showing the differences caused by metformin on the intestinal microbiota

  1. Line 265: MTX alters salivary microbiota more than intestinal – has this been found in humans?  Needs a reference.

Response :

The effect of MTX on oral flora has been reported in humans. As requested, this has been referenced in the new submitted version. Please check the new submitted version lines 321 and 322; page 15.

  1. Implementation of Pharmacomicrobiomics in Clinical Practice: this paragraph needs some re-working as we are not yet in a position to begin recommending specific probiotics, prebiotics, postbiotics or organisms in order to manipulate the gut microbiome to improve drug efficacy/toxicity.

The review is a good summary of data to date – showing that there is definitely an important role for the gut microbiome in the mechanisms for altering drug efficacy/toxicity, but the field is still in its infancy – and exact molecular mechanisms area still elusive. 

We can hypothesize for a day when we may be able to manipulate the gut microbes to assist in improving bioavailability and/or decrease toxicity; however, it is not accurate to suggest that adding or subtracting the specific organisms that have been discussed here is ready for clinical prime time in humans.

Response :

We totally agree and we have made changes to this paragraph, insisting on further studies in this field. Please check the new submitted version lines 356 and 379; page 17.

Indeed, we have not issued firm recommendations as to the use of a particular probiotic or antibiotic with a specific drug, but we want, through this paragraph, to draw attention to the different possible ways to explore for a better consideration of the intestinal microbiota in the management of patients

Minor comments

  1. Line 32: reword sentence; break into 2 sentences

Response :

Done as requested. Please check the new submitted version : line 33 ; page 3.

  1. Line 35: change to “therapeutic drug monitoring”

Response :

Done as requested. Please check the new submitted version : line 35 ; page 3.

  1. Lines 52-55: need references for these statements

Response :

Done as requested. Please check the new submitted version : line 55 ; page 3.

  1. Line 108: “species shows increased growth in diabetic  patients” needs reference

Response :

Done as requested. Please check the new submitted version : line 120 ; page 7.

  1. Line 115: change “conductive” to “conducive”

Response :

Done as requested. Please check the new submitted version : line 129 ; page 7.

  1. Line 139: change to “xenometabolism”
  2. Line 188: change to “intense diarrhea in up to 80%

Response :

Done as requested. Please check the new submitted version : line 212 ; page 11.

  1. Line 213: add “in an animal model” after b-glucosidases

Response :

Done as requested. Please check the new submitted version : line 253 ; page 13.

  1. Line 227: reference the statement: “10% of inter-individual variations of gut flora composition can be explained by drug use.”

Response :

Done as requested. Please check the new submitted version : line 267 ; page 13.

  1. Line 257: change “cancerology” to “cancer”

Response :

Done as requested. Please check the new submitted version : line 313 ; page 15.

  1. Line 273: Rephrase 1st sentence – incomplete sentence

To avoid misunderstanding, we have deleted the sentence. Please check the new submitted version : line 329 ; page 16.

  1. Line 279: change “approaches have not been pursued” to “approaches have nott achieved clinical relevance”

Response :

Done as requested. Please check the new submitted version : lines 334 and 335 ; page 16.

  1. Line 284: consider changing section title to “Treatment failure with TNF alpha inhibitors due to bacteria”

Response :

Done as requested. Please check the new submitted version : lines 340 and 341 ; page 16.

  1. Line 286: (anti-TNFa ) is used…

Response :

Done as requested. Please check the new submitted version : line 342 ; page 16.

Round 2

Reviewer 2 Report

The authors have adequately responded to my comments - thank you.

Minor comment:

line 35:  therapeutic drug monitoring does not need quotation marks around it